# Perceptions of professional soccer coaches, support staff and players toward virtual reality and the factors that modify their intention to use it

**Ben Greenhough** [1,2☯]*, **Steve Barrett** [3‡], **Chris Towlson** [1‡], **Grant Abt** [1‡]

1 Department of Sport, Health and Exercise Science, University of Hull, Kingston upon Hull, United Kingdom,
2 Sports Science and Medicine Department, Hull City AFC, Kingston Upon Hull, United Kingdom,
3 Department of Sports Science and Research Innovation, Playermaker™, London, United Kingdom

☯ These authors contributed equally to this work.
‡ SB, CT and GA also contributed equally to this work.
* b.greenhough-2018@hull.ac.uk

**Data Availability Statement:** https://osf.io/t9j4b/ The majority of data has been available with the exception of some demographic data which has been removed to protect the identity of the

## Abstract

A small evidence base supports the use of virtual reality in professional soccer, yet there is a lack of information available on perceptions and desire to use the technology from those employed at professional soccer clubs. Therefore, the aim of the study was to compare and quantify the perceptions of virtual reality use in soccer, and to model behavioural intentions to use this technology. This study surveyed the perceptions of coaches, support staff, and players in relation to their knowledge, expectations, influences and barriers of using virtual reality via an internet-based questionnaire. To model behavioural intention, modified questions and constructs from the Unified Theory of Acceptance and Use of Technology were used, and the model was analysed through partial least squares structural equation modelling. Respondents represented coaches and support staff (n = 134) and players (n = 64). All respondents generally agreed that virtual reality should be used to improve tactical awareness and cognition, with its use primarily in performance analysis and rehabilitation settings. Generally, coaches and support staff agreed that monetary cost, coach buy-in and limited evidence base were barriers towards its use. In a sub-sample of coaches and support staff without access to virtual reality (n = 123), performance expectancy was the strongest construct in explaining behavioural intention to use virtual reality, followed by facilitating conditions (i.e., barriers) construct which had a negative association with behavioural intention. Virtual reality has the potential to be a valuable technology within professional soccer although several barriers exist that may prevent its widespread use.

## Introduction

Soccer 'performance' is a multifactorial construct comprised of tactical, technical, cognitive and physical components [1–3]. To improve these components (and therefore performance),

respondents of the survey. All data relating to the descriptive statistics and structural equation model are available through the link above.

**Funding:** This work received funding in the form of salary for author SB from Playermaker. The specific roles of this author are articulated in the 'author contributions' section.

**Competing interests:** BG is currently affiliated with a virtual reality company that work with professional soccer clubs. However, he was not affiliated with that company during the work reported in this study, and the company played no role in the study. SB is affiliated with Playermaker, and received funding from them in the form of salary. There are no patents, products in development or marketed products associated with this research to declare. This does not alter our adherence to PLOS ONE policies on sharing data and materials.

information technology systems such as athlete tracking devices that include global positioning systems (GPS) have been utilised to permit greater understanding of the physical demands imposed on soccer players during training and match-play, ultimately enhancing training prescription [4, 5]. However, despite cognitive characteristics of soccer players being considered of equal importance [2], evidence for application of technology (e.g. virtual reality, 2D reality etc) to assess key cognitive attributes such as decision making in soccer are limited [6]. Despite limited research, the application of virtual reality (VR) in professional sport is becoming increasingly common [7], largely due to it becoming more commercially available [8]. Virtual reality is defined as a computer-simulated environment that aims to induce a sense of being mentally and/or physically present in another place [7, 9]. To date, a small evidence base supports its use as a technology to enhance athletic performance. For example, transfer of perceptual motor-skills learned within the virtual world to the real-world has been demonstrated, illustrating the effectiveness of VR as a training tool [10–12]. In addition, VR has also been shown to be effective in tactical analysis sessions with Australian football umpires and varsity basketball players to improve decision-making when compared to conventional methods such as video recordings viewed through a computer screen [13, 14]. As such, VR presents multiple opportunities to be used within professional soccer.

While there appears to be scientific evidence and a supportive rationale for the use of VR in professional soccer, we also need to explore user perceptions of VR to understand if there is a desire to use the technology. This is important because there is more to technology adoption than just technology efficacy. The diffusion of innovation theory suggests that the rate of technology adoption depends more on the end-users' subjective perceptions of usability, complexity and observability of the technology rather than objective evidence for its efficacy [15, 16]. To that end, only one study has explored the perceptions of VR use in professional soccer. Thatcher et al. [8] identified key opportunities for VR implementation including its use during rehabilitation and for youth development, as well as barriers to its use, such as a lack of empirical evidence and the practicality of using it. However, the opinions of soccer coaches interviewed by Thatcher et al. [8] varied markedly on the efficacy of VR, and more importantly the aims of the study did not address the mechanistic factors that influence coaches' intentions to use the technology. In order to understand intention, studies on technology acceptance typically refer to theoretical models from the social sciences specifically designed to examine behavioural intention to use technology [17]. For example, the Unified Theory of Acceptance and Use of Technology (UTAUT) helps to explain how the intent to use a technology is derived from beliefs that the technology will enhance work related performance, that the technology is easy to use, and is manifested from socially orientated influences [18].

Although the research by Thatcher et al. [8] provides valuable information on the facilitators and barriers to VR adoption from the perspective of coaching staff, understanding the perceptions of players is of equal importance as they are a key stakeholder in the adoption of a new technology [19]. Consequently, if the perceptions between those who implement the technology (coaches and support staff) differs to those who will use the technology (players), then it is possible that buy-in to the use of VR in professional soccer could be limited. In addition, although the literature is clear on some of the key opportunities and barriers to VR use, it is unclear how they compare to each another. For instance, it is unclear whether lack of evidence is more of a barrier than cost.

The first aim of this study was to compare the perspectives of those responsible for implementing VR (coaches and support staff) with those who would use VR (players). Second, we aimed to quantify these perspectives to understand how each facilitator or barrier to VR use compare to one another. Third, we aimed to model behavioural intention to use VR in professional soccer coaches and support staff who do not currently have access to this technology.

## Methods

### Survey design and distribution

A cross-sectional survey of coaches, support staff and players working in professional soccer was conducted between December 2019 and April 2020. The study received approval (FHS203) from the Faculty of Health Sciences Research Ethics Committee at The University of Hull.

Due to the nature with which responders interact with VR, two separate surveys were designed to be completed by coaches and support staff (collectively referred to as practitioners from now on), and players, respectively. Both surveys included the following sections: (1) general information; (2) technology acceptance (3) knowledge of VR; (4) performance expectancy; (5) social influence; and (6) barriers to using VR. For the technology acceptance section, respondents completed the 10-question version of the Technology Readiness Index (TRI) 2.0 [20]. The TRI was used to assess respondents' technology readiness, which is defined as a persons' propensity to embrace new technology. Depending on whether responders had access to VR or not, a further section asked questions related to either (7) use of VR, or (8) intention to use VR. Except for five questions requiring a binary response (yes/no), all questions were multiple choice or a Likert scale. For Likert scale questions, four to seven response label anchors were used. Fully labelled Likert scales were used instead of partially labelled scales due to their improved validity and reliability [21]. The Likert scale response labels reflected the relevant constructs of each section (e.g., agreement, influence, barriers) and each section was defined precisely [21].

Depending on the respondents' answers, the surveys consisted of 54–63 questions for practitioners, and 33–39 questions for players. The number of questions completed depended on the respondents' knowledge of VR and if they currently used it within their club. Although some questions were the same in both surveys to allow comparisons, other questions were specific to each group. Inclusion criteria questions were placed at the start of the survey after reading the information sheet and providing informed consent. The inclusion criteria required participants to be 17 years of age or older and have not submitted the survey previously. Additionally, respondents had to be working for a professional soccer club or national association team (practitioner survey) or be in a professional contract at a professional soccer club (players survey).

After completing the general information and technology acceptance sections, VR was defined as, "including a headset worn by a user that covers their eyes, allowing them to experience a virtual world that is created by a computer". As such, our definition did not include other immersive VR modalities such as CAVE systems [22]. To clarify our definition, VR was also defined as "not referring to non-immersive technology that results in an output through a television or other electronic interface, and does not include augmented reality, whereby computer-generated images are placed into the real world and viewed live". Images were included alongside the definitions so that respondents were aware of the type of VR being referred to. To define the context in which questions should be answered, the definition "*Virtual reality used by coaches, support staff and players as part of training or personal use within professional football training grounds*" was used. This was to differentiate from answering in other contexts where VR is used in soccer such as fan engagement [23].

The survey was distributed electronically via email to contacts known by the research team. For clubs where no relationship existed with the research team, an invitation email was sent to the Head of Medicine and Sport Science, or the equivalent position. Within the invitation email, information was provided on the aims and benefits of the research. Recipients were asked to circulate the surveys in their club to appropriate practitioners (technical coaching

staff, staff working in medical or performance, performance analysts) and players. A reminder email was sent out one month before the closing date. Additionally, the survey was circulated openly on the social media platform, Twitter. Data were collected using an online survey platform (www.onlinesurveys.ac.uk, Bristol, UK), with links to both surveys included in the email or on social media. The practitioner and player surveys took approximately 10 and 6 minutes to complete, respectively. As respondents were encouraged to share the survey with those in their team, and the surveys were circulated openly through social media, it was not possible to determine the response rate to the surveys.

## The partial least squares structural equation model specification

Data were analysed using a partial least square structural equation model (PLS-SEM). Specification of the PLS-SEM model involved the development of inner and outer models. The inner model involved specification of the path models between the independent and dependent construct variables. The outer model was specified by connecting the indicator variables that corresponded with the constructs specified in the inner model. The indicator variables that corresponded to the constructs were created by the lead researcher and were based on theoretical knowledge in the literature.

In developing the PLS-SEM, we modified the UTAUT constructs as devised by Venkatesh et al. [18] to be appropriate for this study. As such, the performance expectancy, social influence and facilitating conditions constructs were used, under new definitions. Performance expectancy was defined as "the degree of belief that virtual reality will improve performance", social influence as "the degree of being socially influenced to use virtual reality" and facilitating conditions as "the degree that barriers are in place to use virtual reality". Effort expectancy was not included in this study under the assumption that most participants would not have access to VR. Performance expectancy and social influence consisted of five indicator variables, whereas facilitating conditions consisted of seven indicator variables (Table 1). All the adapted UTAUT constructs were specified as formative constructs. Technology readiness was included as a single item construct using the overall score of the TRI 2.0 questionnaire which was calculated through methods outlined previously [20]. Likeliness to use VR was specified as the dependent construct variable, a reflective construct and consisted of two indicator variables (Table 1).

## Sample size calculation

An *a priori* sample size calculation was conducted for the PLS-SEM. Because PLS-SEM builds on ordinary least squares regression, statistical power analyses for multiple regression models will result in a satisfactory sample size estimation [24]. As such, sample size estimations were carried out using G$^*$Power software [25], using the F tests, linear multiple regression: Fixed model, $R^2$ deviation from zero option. Sample size was estimated to detect a moderate effect size ($f^2 = 0.15$), using a statistical power of 0.9 and with alpha set to 0.05. A moderate effect size was chosen based on the findings of a similar study that have used the UTAUT model [17]. Seven predictors were chosen based on the number of formative indicators required for the facilitating conditions construct [26] (Table 1). From these inputs a minimum sample size of 130 participants was required.

## Data analysis

Data were analysed in two stages–a suitable descriptive analysis and then construction of the PLS-SEM model [27]. For categorical, multiple choice and Likert scale questions, frequency analysis was conducted with percentages and number of participants reported. Data were

**Table 1. Specification of the PLS-SEM constructs, indicator variables and survey questions.**

| Construct type | Construct | Indicator variable | Question |
|---|---|---|---|
| Independent | Performance expectancy | Physical | Physical fitness (i.e., Virtual reality used with players to improve areas such as strength, power, aerobic fitness etc.) |
| | | Cognition | Cognition (i.e., Virtual reality used with players to improve cognition such as decision making, reaction time, visual awareness etc.) |
| | | Technical | Technical skill (i.e., Virtual reality used with players to improve technical ability such as passing & shooting accuracy etc.) |
| | | Tactical | Tactical development (i.e., Virtual reality used with players to improve awareness of team tactics etc.) |
| | | Mental Wellbeing | Mental wellbeing (i.e., Virtual reality used with players to improve mental wellbeing such as stress and anxiety etc.) |
| Independent | Social influence | To be seen using | To be seen using an innovative technology (i.e., I would be influenced to use virtual reality so that others see me using an innovative technology) |
| | | Influential others | Influential others use virtual reality (i.e., I would be influenced to use virtual reality if individuals that influence me also use it) |
| | | Influential clubs | Influential clubs use virtual reality (i.e., I would be influenced to use virtual reality if clubs that influence me also use it) |
| | | Seniors want it used | Those senior to me (i.e., I would be influenced to use virtual reality if individuals that are senior to me want it to be used) |
| | | Players enjoy using | Player enjoyment (i.e., I would be influenced to use virtual reality if players enjoyed using the system) |
| Independent | Facilitating conditions | Player buy in | Player buy-in (i.e., Getting players to engage with the virtual reality system is a barrier to using it) |
| | | Coach buy in | Coach and support staff buy-in (i.e., Coaching staff buy-in to virtual reality being used with players is a barrier to using it) |
| | | Space to operate | Personnel to operate (i.e., requiring personnel to operate the virtual reality system is a barrier to using it) |
| | | Personnel to operate | Space to operate (i.e., space within the training ground to operate the virtual reality system is a barrier to using it) |
| | | Limited evidence | Limited evidence base (i.e., limited research available on virtual reality used in professional football is a barrier to using it) |
| | | Time available | Time available (i.e., time available to use within schedule is a barrier to using virtual reality) |
| | | First impression | First impression (i.e., my first impression of using, seeing, or hearing about virtual reality is a barrier to using it) |
| Independent | Technology readiness | TRI 2.0 overall score | Overall score of the 10 technology readiness questions |
| Dependent | Likeliness to use | Intention | If virtual reality technology was made available to you, how likely are you to use it within your club? |
| | | Opinion | What is your overall opinion of virtual reality technology for use by coaches, support staff and players within the training ground setting? |

confirmed as being not normally distributed using the Shapiro-Wilk test and quartile-to-quartile plots. However, PLS-SEM is a nonparametric statistical method, so normality is not a required assumption [28].

The PLS-SEM was assessed through the evaluation of inner and outer models. As formatively and reflectively measured constructs are based on different concepts, different evaluation measures took place [27]. In the assessment of the reflective construct, evaluation of the internal consistency reliability, convergent validity and discriminant validity took place. To assess the internal consistency reliability, Cronbach's alpha and composite reliability were calculated, with scores above 0.7 considered as satisfactory [29]. Convergent validity was assessed, which is the extent that an indicator correlates positively with alternative indicators of the same construct. For reflective constructs, outer loadings should be above 0.7, with mean variance extracted value calculated as the grand mean value of the indicator loadings associated with the construct [29]. A value of 0.5 and above is considered satisfactory and indicates that a

construct explains more than half the variance of its indicators. Discriminant validity was also assessed and was defined as the extent that the construct is truly distinct from other constructs [29]. First, the indicator's outer loadings should be higher than all its cross-loadings of other constructs. Secondly, the Fornell-Larcker criterion should demonstrate that the square root of the mean variance extracted in the intention to use construct is higher than its highest correlation with the other constructs within the model.

To assess the formative constructs, evaluation of collinearity, compatibility of the data with the hypothesis, and relevance of the formative indicators took place. For collinearity, variance inflation factor (VIF) of each indicator was assessed, with VIF above 5 indicating potential collinearity, and the ideal threshold set to 3.3 and below. A bootstrapping procedure (bias-corrected and accelerated) with 5000 resamples was carried out to determine statistical compatibility of the data with the hypothesis (evaluation of the p value relative to the *a priori* alpha of 0.05) and relevance of the indicators [30]. Where the p value for an indicator's weight was above the *a priori* alpha, the indicator's absolute contribution to the construct was considered via assessing its outer loading. That is, the correlation between the indicator variable and the construct when no other indicators are taken into consideration. Indicators with weights and outer loadings incompatible with the alternative hypothesis, were removed from the construct as the indicator provided no meaningful explanation in forming the construct. Indicator variables with a weight that was incompatible with the alternative hypothesis but had outer loading above .5 and compatible with the alternative hypothesis, remained in the construct. However, indicators with outer loadings below .5 and compatible with the alternative hypothesis, required (1) a rationale for its inclusion in the formative construct model through either anecdotal or empirical evidence [29], and (2) removing the indicator variable doesn't change the conceptual meaning of the construct [31].

For the inner model, collinearity between constructs was assessed using the VIF criteria as outlined previously. The inner model was then assessed by means of (1) the size and statistical relevance of the path coefficients ($\beta$); (2) the explained variance ($R^2$); and (3) the path coefficient effect size ($f^2$) [29]. The $R^2$ value represented the independent construct variables combined effect on the dependent construct variable. The $R^2$ values of 0.25 'weak', 0.5 'moderate' and 0.75 'substantial' were used. Effect sizes ($f^2$) were reported as 0.02 'small', 0.15 'medium' and 0.35 'large', and represented the independent construct variables contribution to the dependent construct variable $R^2$ value [32]. Descriptive statistics were generated using R Studio [33], with figures produced using the 'Likert' package [34]. The multivariate model was analysed using SmartPLS 3.0 [35]. The *a priori* alpha was set at $p < 0.05$. To evaluate p values derived from the PLS-SEM, we adopt the recommendation of Greenland et al. [36] where each p value is evaluated as a measure of the degree of statistical compatibility between an hypothesis and the data (given a model used to generate it), bounded by 0 (complete incompatibility, data impossible under the hypothesis and model) and 1 (no incompatibility apparent from the test). Reported p values $> 0.05$ are therefore evaluated as being compatible with the null hypothesis and p values $< 0.05$ are evaluated as compatible with the alternative hypothesis.

## Results

### Respondent demographics

Overall, 207 respondent completed the survey (practitioners: n = 143; players: n = 64). Practitioners worked in roles that included sport scientists (n = 46), physiotherapists (n = 26), performance analysts (n = 25), strength and conditioning coaches (n = 19), technical coaches (n = 13), head coach/manager (n = 6) and other roles (n = 8) such as heads of performance, rehabilitation coaches and heads of innovation and research. Respondent demographics are

**Table 2. Proportion and frequency of respondent demographics for practitioners and players.**

| Demographic | Characteristics | Descriptive statistics | | PLS-SEM [a] |
| --- | --- | --- | --- | --- |
| | | Practitioner % (n) | Player % (n) | Practitioner % (n) |
| Gender | Male | 94% (135) | 98% (63) | 95% (117) |
| | Female | 6% (8) | 2% (1) | 5% (6) |
| Age | 17–21 | 1% (2) | 53% (34) | 2% (2) |
| | 22–26 | 21% (30) | 23% (15) | 19% (23) |
| | 27–31 | 36% (51) | 17% (11) | 38% (47) |
| | 32–36 | 15% (22) | 5% (3) | 16% (20) |
| | 37–41 | 14% (20) | 2% (1) | 15% (19) |
| | 42–46 | 4% (6) | | 4% (5) |
| | 47–51 | 6% (8) | | 3% (4) |
| | 52+ | 3% (4) | | 2% (3) |
| Tier | Tier 1 | 45% (64) | 6% (4) | 46% (57) |
| | Tier 2 | 38% (54) | 72% (46) | 34% (42) |
| | Tier 3 | 6% (9) | 9% (6) | 7% (8) |
| | Tier 4 | 6% (8) | 13 (8) | 7% (8) |
| | Tier 5 | 1% (2) | | 2% (2) |
| | National association team | 4% (6) | | 5% (6) |
| Team | Senior players | 61% (87) | | 57% (70) |
| | Senior academy players | 27% (39) | | 30% (37) |
| | Academy players | 11% (16) | | 12% (15) |
| | Junior academy players | 1% (1) | | 1% (1) |
| Country | England | 62% (89) | 94% (60) | 59% (73) |
| | United states of America | 8% (12) | | 8% (10) |
| | Scotland | 6% (9) | | 7% (9) |
| | Australia | 7% (10) | 1% (1) | 8% (10) |
| | Other | 17% (23) | 5% (3) | 18% (21) |
| Working gender | Male | 94% (135) | | 93% (115) |
| | Female | 6% (8) | | 7% (8) |
| Highest qualification | PhD | 9% (13) | | 10% (12) |
| | Masters | 55% (78) | | 57% (70) |
| | Bachelors | 28% (40) | | 25% (31) |
| | Other | 8% (12) | | 8% (10) |

Respondent demographics are included for the descriptive statistics and the PLS-SEM.

[a] PLS-SEM–Partial least squares structural equation model.

displayed in Table 2. For the PLS-SEM, 123 practitioners were included based on them currently having no access to VR. Respondent demographics for this sub-group are available in Table 2.

## Descriptive statistics

**Awareness and experience of VR.** Most practitioners (94%) and players (89%) knew what VR was, based on the definitions and images provided in the survey. Additionally, most practitioners (76%) and players (72%) were aware of VR being used within professional soccer training grounds. Most practitioners (70%) and players (54%) had never used VR within a professional training ground, however, those that had used VR within a professional training ground did so within the last year (practitioners: 22%; players: 44%) (S1 Table).

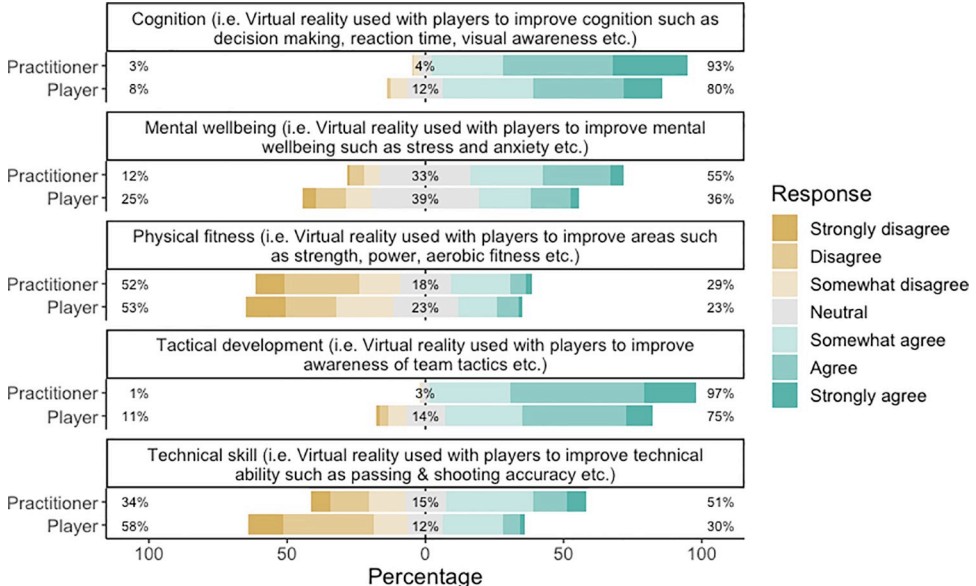

**Fig 1. Responses by practitioners and players to statements regarding what VR should be used for.** Percentages indicate overall disagreement, neutral and overall agreement, from left to right respectively.

**Performance expectancy.** Practitioners and players responded similarly regarding how they perceived VR could improve performance (Fig 1). Responses by practitioners and players indicated in favour of agreement that VR could improve cognition (practitioner: 93%; player: 80%) and tactical performance (practitioner: 97%; player: 75%). Similarly, approximately half of practitioners (52%) and players (53%) indicated in favour of disagreement that VR could improve physical performance. Practitioners generally agreed that VR should be used for performance analysis (93%), followed by preparation (77%) and rehabilitation (73%), whereas there was no consensus on whether VR should be used for player monitoring or talent identification (Fig 2).

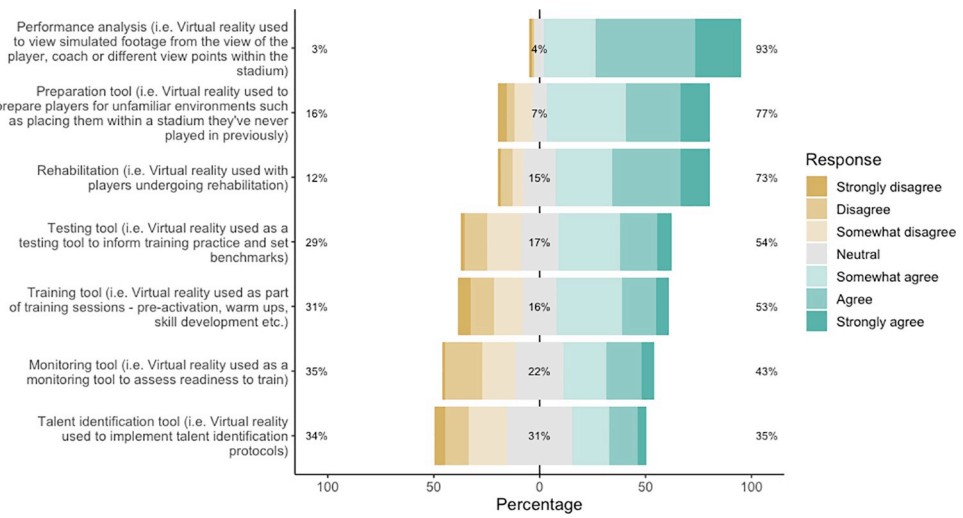

**Fig 2. Responses by practitioners to statements regarding how VR should be used.** Percentages indicate overall disagreement, neutral and overall agreement, from left to right respectively.

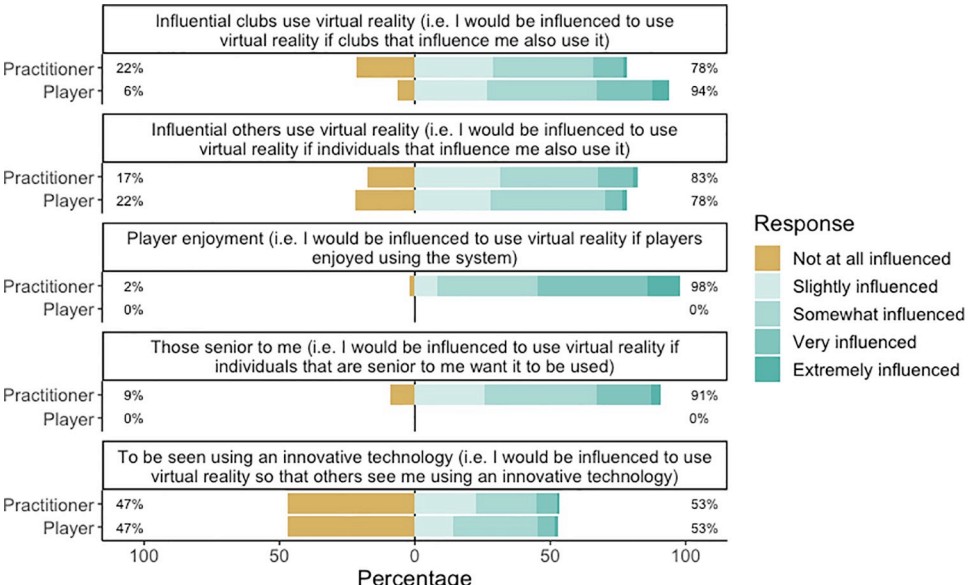

**Fig 3. Responses by practitioners and players to statements on how they perceive they are influenced to use VR.** Percentages indicate not at all influenced, and overall influenced, from left to right respectively.

**Social influence.** Practitioners and players responded similarly on influences to use VR (Fig 3). Practitioners and players are influenced to use VR in some capacity if influential clubs use VR (practitioners = 78%, players = 98%) and if influential others use VR (practitioners = 83%, players = 78%). Additionally, practitioners responded as being very influenced to use VR if players enjoyed using it (98% overall influence) and somewhat influenced if those senior to the player wanted it to be used (91% overall influence).

**Facilitating conditions.** Of the barriers that practitioners responded to, monetary cost was rated as the largest barrier, albeit in a reduced sample of 36 respondents who were aware of the associated cost of VR. As such, 107 participants were not aware of the costs associated with VR. Limited research within football and time available to use VR were also rated as moderate barriers to using the technology. First impression of VR was rated as the lowest barrier, with 45% of respondents indicating it as not being a barrier (Fig 4).

**Opinion of VR.** Practitioners and players responded similarly on their overall opinion of VR being used within soccer training ground facilities (Fig 5). In both groups, half of respondents generally viewed VR as being positive, whereas 13% and 11% of practitioners and players, respectively, indicated negative opinions of VR.

## Multivariate analysis: Factors to determine acceptance of VR if given access

**Model validity and reliability: Outer model assessment.** The results of the reflective constructs internal consistency reliability, convergent validity and discriminant validity are shown in Table 3. Cronbach alpha and composite reliability were above 0.7, indicating satisfactory reliability [37], outer loadings were above 0.7, and the mean variance extracted was above 0.5, demonstrating satisfactory convergent validity. Fornell-Larcker criterion showed that the square root of the mean variance extracted was greater than the related inter-construct correlations of the other constructs in the model, illustrating that adequate discriminant validity existed.

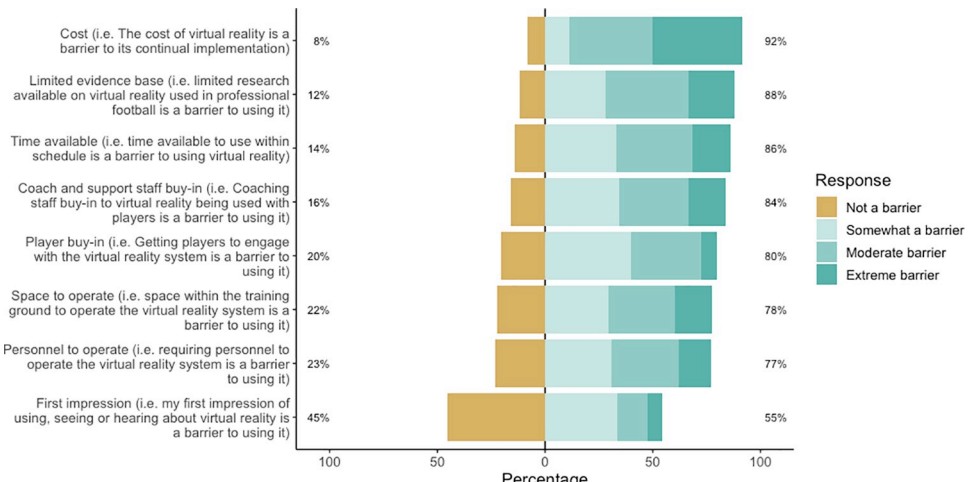

**Fig 4. Responses by practitioners to statements regarding facilitating conditions to using VR.** Percentages indicate not a barrier and overall barrier, from left to right respectively. For the statement 'cost', only 36 respondents responded as being aware of the monetary cost associated with VR.

The VIF illustrated that all indicators were below the optimal threshold of 3.3, indicating no collinearity issues existed and therefore multicollinearity was not an issue for estimating the PLS path model (Table 4).

Each construct's indicator weight and loadings are outlined in Table 4. For the performance expectancy construct, the indicator weights were compatible with the alternative hypothesis for cognition and physical ($p = < 0.001$–0.018). Indicator weights for technical, tactical, and mental wellbeing were compatible with the null hypothesis ($p = 0.485$–0.733), however outer loadings illustrated contributions to the overall construct compatible with the alternative hypothesis. For the social influence construct, indicator weights were compatible with the alternative hypothesis for influential clubs and player enjoyment ($p = 0.002$–0.026), whereas to be seen using, influential others, and seniors want it used were compatible with the null hypothesis ($p = 0.633$–0.979). However, the outer loading illustrated a contribution to the construct compatible with the alternative hypothesis ($p = < 0.001$–0.01), and therefore remained.

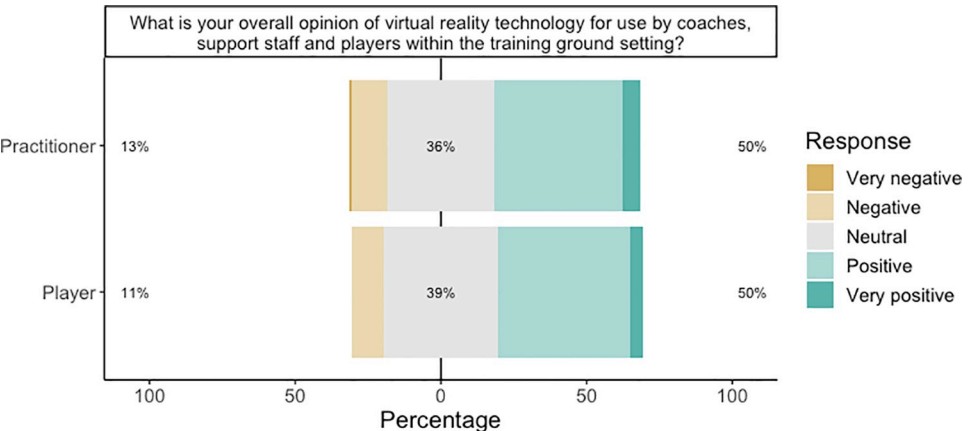

**Fig 5. Responses by practitioners and players to a statement regarding the overall opinion of VR being used within the training ground setting.** Percentages indicate overall negativity, neutral and overall positivity, from left to right respectively.

**Table 3. Internal consistency reliability, convergent validity and discriminant validity of the reflective construct variable, likeliness to use.**

| Constructs | Outer loadings | Cronbach Alpha | Composite reliability | Average variance extracted | Fornell-Larcker criterion |
| --- | --- | --- | --- | --- | --- |
| | | | | | Likeliness to use |
| Likeliness to use | (0.914, 0.904) | 0.79 | 0.905 | 0.826 | 0.909 |
| Performance expectancy | | | | | 0.620 |
| Social influence | | | | | 0.376 |
| Facilitating conditions | | | | | -0.502 |

Finally, facilitating conditions constructs indicator weights were compatible with the alternative hypothesis for coach buy in and limited evidence base (p = 0.001–0.011). The remaining five indicators were compatible with the null hypothesis (p = 0.062–0.915). On assessment of the outer loadings, all displayed contributions to the overall construct compatible with the alternative hypothesis, except for space to operate (p = 0.503) which illustrated no absolute contribution to the overall construct. As such, the space to operate indicator was removed from the construct. Finally, multicollinearity of the independent construct variables was assessed through the VIF. All VIF values were below the optimal VIF value of 3.3, indicating no issues of collinearity between the independent construct variables and the dependent construct variable (Table 4). Technology acceptance (single item construct) had a VIF of 1.005, indicating no issue of collinearity.

**Inner model assessment.** The PLS-SEM results displayed in Table 5 illustrate that the path from performance expectancy to Likeliness to use was positive ($\beta$ = .465, $t$ = 7.028,

**Table 4. Formative construct indicators item weight and outer loading.** Also included are the item weight and outer loading t statistic and p value, and the item weight 95% bias-corrected and accelerated confidence interval.

| Formative construct | Indicator | Item weight (outer loading) | Item weight t. stat (p value) | 95% Bca confidence interval | Outer loading t.stat (p value) | VIF | Full collinearity |
| --- | --- | --- | --- | --- | --- | --- | --- |
| Performance expectancy | Physical | 0.326 (0.635) | 2.372 (0.018) | 0.044,0.579 | 5.965 (0.000) | 1.283 | |
| | Cognition | 0.66 (0.847) | 5.016 (0.000) | 0.404, 0.917 | 10.452 (0.000) | 1.647 | |
| | Technical | 0.33 (0.733) | 1.867 (0.062) | 0.012, 0.692 | 3.688 (0.000) | 1.607 | 1.326 |
| | Tactical | -0.083 (0.485) | 0.575 (0.565) | -0.37, 0.184 | 7.306 (0.000) | 1.542 | |
| | Mental Wellbeing | 0.062 (0.52) | 0.411 (0.681) | -0.221,0.361 | 4.532 (0.000) | 1.306 | |
| Social influence | Seen using | 0.008 (0.525) | 0.026 (0.979) | -0.539, 0.635 | 2.754 (0.006) | 1.646 | |
| | Influential others | -0.044 (0.681) | 0.106 (0.915) | -0.89,0.713 | 3.935 (0.000) | 3.043 | |
| | Influential clubs | 0.736 (0.802) | 2.224 (0.026) | 0.131, 1.392 | 5.55 (0.000) | 2.928 | 1.252 |
| | Seniors want it used | -0.156 (0.563) | 0.478 (0.633) | -0.795,0.461 | 2.586 (0.01) | 1.777 | |
| | Players enjoy using | 0.671 (0.78) | 3.068 (0.002) | 0.247,1.035 | 4.429 (0.000) | 1.374 | |
| Facilitating conditions | Player buy in | 0.022 (0.376) | 0.106 (0.915) | -0.365,0.452 | 2.217 (0.027) | 1.298 | |
| | Coach buy in | 0.496 (0.722) | 2.559 (0.011) | 0.13, 0.86 | 6.065 (0.000) | 1.431 | |
| | Space to operate | -0.15 (0.115) | 0.89 (0.373) | -0.478,0.178 | 0.67 (0.503) | 1.177 | |
| | Personnel to operate | 0.047 (0.381) | 0.278 (0.781) | -0.287, 0.372 | 2.318 (0.021) | 1.241 | 1.076 |
| | Limited evidence | 0.56 (0.728) | 3.387 (0.001) | 0.256, 0.897 | 6.586 (0.000) | 1.339 | |
| | Time available | -0.031 (0.451) | 0.19 (0.849) | -0.344,0.308 | 3.227 (0.001) | 1.489 | |
| | First impression | 0.348 (0.686) | 1.865 (0.062) | -0.034, 0.691 | 5.093 (0.000) | 1.453 | |

Bca–Bias-corrected and accelerated confidence interval.

**Table 5. Path coefficients between the independent and dependent construct variables.** Also included are the t-value, p-value, effect size, and the dependent construct variables explained variance.

| Path | Path coefficient β (95% CI) | t-value | p-value | Effect size $f^2$ (95% CI) | Explained variance $R^2$ |
|---|---|---|---|---|---|
| Performance expectancy—Likeliness to use | 0.465 | 7.028 | < 0.001 | 0.343 | .523 |
| | (0.336, 0.592) | | | (0.173,0.674) | |
| Social influence—Likeliness to use | 0.131 | 1.924 | 0.054 | 0.029 | |
| | (-0.042, 0.231) | | | (0.002,0.175) | |
| Facilitating conditions—Likeliness to use | -0.364 | 5.164 | < 0.001 | 0.259 | |
| | (-0.489, -0.209) | | | (0.099,0.591) | |
| Technology acceptance—Likeliness to use | 0.039 | 0.617 | 0.537 | 0.003 | |
| | (-0.082, 0.162) | | | (0,0.059) | |

p = <0.001, $f^2$ = 0.34), whereas facilitating conditions was negative (β = -0.364, $t$ = 5.164, p = <0.001, $f^2$ = 0.26). The path coefficients of social influence (β = .131, $t$ = 1.924, p = 0.054, $f^2$ = 0.029) and technology acceptance (β = 0.039, $t$ = 0.617, p = 0.537, $f^2$ = 0.003) were compatible with the null hypothesis. The explained variance ($R^2$) in the dependent construct by the independent constructs was .523, indicating moderate in-sample explanatory power of the structural model (Table 5).

## Discussion

The main findings of our study were: (1) the PLS-SEM model indicated that performance expectancy positively contributed towards likeliness to use VR and was the strongest overall contributor to the model (Table 5), (2) the second largest contributor to the model was facilitating conditions, which had a negative relationship with likeliness to use (Table 5), (3) the model indicated 'first impression' as a significant absolute contributor to the facilitating conditions construct (Table 4), (4) social influence and technology acceptance did not contribute towards likeliness to use VR (Fig 6), and (5) practitioners generally agreed VR should be used in performance analysis and rehabilitation (Fig 2).

Performance expectancy, in most cases, has been reported as the primary determinant of intention to use a technology [18], and our findings are consistent with that of Liu et al. [17]. This comes as no surprise given that modern technology has had a substantial impact on professional sport, with many practitioners considering advances in technology to be invaluable [38]. The second largest contributor towards likeliness to use VR was facilitating conditions (e.g., limited evidence base, coach buy-in). While there is a growing evidence base to support the efficacy of VR in sport, its applications for improving soccer performance are not available. As such, this may lead to early scepticism of VRs value, creating beliefs that the technology is a 'gimmick' or 'novelty' [8]. Although half of the respondents indicated first impression as not being a barrier towards VR use (Fig 4), our model indicated first impression as an absolute contributor to the facilitating conditions construct (Table 4). First impression bias refers to a limitation in human information processing whereby individuals are strongly influenced by the first piece of information they receive, and that future information is biasedly evaluated to fit the narrative of the original information [39]. This finding is interesting, as the results may suggest that there is initial scepticism towards the technology without deeper considerations of how it could be beneficial within the training ground environment.

Social influence and technology readiness had small ($f^2$ = 0.03) and no effect ($f^2$ = 0.003) on likeliness to use VR, respectively (Fig 6). This finding is consistent with a previous study reporting that social influence and technology readiness did not contribute to the behavioural

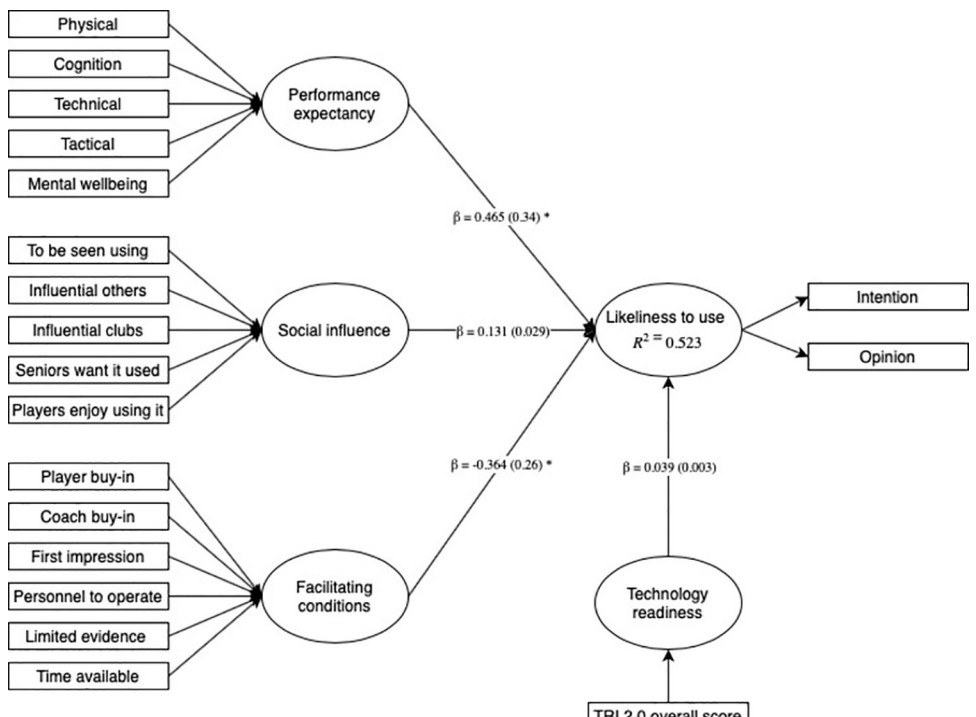

**Fig 6. Node diagram showing the path coefficients between the independent construct variables and the dependent construct variables.** β = beta coefficients; ($f^2$) = path coefficient effect size; R2 = explained variance. * statistically significant at < 0.001.

intentions of recreational golfers to use smart technology, such as VR and GPS devices [40]. Further, social influence has been reported to be a non-contributing factor in the intention to use new technologies for rehabilitation in physical and occupational therapists [17]. One possible reason for social influence not being a contributing factor in VR adoption is the voluntary context in which respondents anticipated they would use this technology. Previous research has shown that when technology use is explored within voluntary contexts, its contribution to technology acceptance is limited [18].

The current study also aimed to understand respondents' perceptions of how VR should be used within professional soccer. These perceptions were not included as part of the model but used to further understand the direction of VR within the sport. Practitioners generally agreed that VR should be used for performance analysis purposes, followed by using it as a preparation tool (visiting new environments, i.e., stadiums) and for rehabilitation (Fig 2). Regarding rehabilitation, VR may provide players with the opportunity to continue to train mentally in the absence of outside soccer-specific training. This could be achieved through soccer-specific drills using a virtual ball, thereby eliminating physical contact. Alternatively, those in the early stage of rehabilitation where movement is not possible (such as post-surgery), match footage could be revisited while viewing the game from their own perspective. Our results support the views expressed previously by coaches in professional soccer, who have indicated the value in VR being used during player rehabilitation [8]. Further, this finding is supported by research showing the potential of VRs ability to promote a dissociative attentional focus, distracting the user from the exercise performed [41] which in the case of the study by Gokeler et al. [42] resulted in patients with anterior cruciate ligament reconstruction having a greater movement proficiency while using VR, compared to not using VR.

Generally, practitioners and players shared a similar overall opinion of VR, with very few indicating negative opinions of the technology (Fig 5), although a high percentage of neutrality was evident. However, the low frequency of negative opinions is encouraging for the future use and adoption of VR in soccer. That said, it is important to note that in the practitioners' sample, 86% of respondents had no access to VR in their clubs, and 70% had never used a soccer-specific VR system. Their limited experience with this technology, despite half indicating positive views is interesting. One could look towards models such as the Gartner hype cycle to help explain this. The Gartner hype cycle model, developed by Gartner Inc, is used to explain the generalised evolutionary path that technology takes over time [43]. In the early stages when adoption rates are low, an overly positive reaction to the technology is seen through people's attraction to novelty, social contagion and unclear attitudes towards decision making [44]. However, once the period of 'hype' and overenthusiasm has passed, organisations and users of the technology experience dissatisfactory results that don't match with their expectations, causing some users to abandon the technology (so called 'trough of disillusionment'). Further investments in the technology allow for greater applications, knowledge, and socialisation ('slope of enlightenment'), which finally leads the technology to be realistically valued in the marketplace, and adoption begins to accelerate ('plateau of productivity'). As of 2016, VR was recognised on the Gartner hype cycle within the slope of enlightenment [45], indicating that VR had begun to find its place in the market. However, this has been largely due to the gamification of VR which differs to how it's used in soccer. Virtual reality in soccer might therefore be in an early stage of 'hype' where adoption rates are low, and its future success still unclear.

## Limitations

In the present study, it is important to note that although the questions included in our survey to form each construct were based on theoretical evidence and observations of VR use in professional soccer (and other fields), our constructs may not fully represent what they are trying to convey. For example, in the social influence construct we did not include a question on the influence of technology companies, who may provide bold claims on the benefits associated with VR [46]. This inclusion as an example may have revealed the construct as a larger contributor towards likeliness to use VR. Further, the survey was made available to anyone working within a professional soccer club, irrespective of how many respondents may have come from the same club. This is in contrast to previous survey based research, where only one respondent was permitted to complete the survey in order to reduce respondent bias [47]. While it was always the intention to understand individual perceptions, it is possible that if respondents came from the same club and shared a club based philosophy on the use of VR, then this may have inflated the results in a given direction [8]. Finally, the timing of the COVID-19 global pandemic caused most soccer leagues to suspend matches with some countries imposing nationwide lockdowns, preventing training ground access. During this time, we continued to collect data and it's possible that the respondents perception of VR may have been influenced because of a necessity to train in isolation, a tool in which VR has been recommended [48].

## Conclusions

Virtual reality is still a relatively new technology that has been adopted by a small number of professional soccer clubs. Prior to this study, little was known about the current perceptions, influences and barriers of VR and their contribution to future VR adoption. Our study allows us to conclude that likeliness to use VR in professional soccer mostly depends on expectations concerning the performance benefits of using the technology. In other words, the belief that

VR would improve soccer-related performance was the most important factor in determining likeliness to use VR. Additionally, likeliness to use VR did not depend on socially orientated influences, or the general tendency to accept new technologies. However, our results revealed that likeliness to use VR depends on the belief that there are barriers facilitating the use of VR within the training ground environment. In other words, the greater a person believes that barriers are in place to using VR, the less likely they are to use the technology.

## Practical applications

Practitioners in soccer clubs who are looking to implement VR need to work closely with multiple key stakeholders, providing them with evidence of how VR could support their practice. For instance, physiotherapists and rehabilitation coaches should be educated on the use of VR to improve movement proficiency with injured players [42], whereas performance analysis departments and technical soccer coaches should be informed of how VR could be used during tactical analysis sessions to increase engagement among players [13]. In contrast, focusing on the social influences of VR in soccer (e.g., which clubs are using the technology) is not the right approach when discussing VR with stakeholders. Practitioners should also investigate ways of alleviating key barriers of VR adoption in soccer. For instance, collaborating with universities who have access to VR could work effectively if the technology is loaned to soccer clubs in exchange for the opportunity to conduct VR research within the club environment, thereby eliminating the cost barrier and increasing the available research evidence.

## Supporting information

**S1 Fig. Practitioners' perceptions of which age-group virtual reality should be used with.** S1 Fig. Likert bar-plot of responses by coaches/support staff to statements regarding who VR should be used with. Percentages indicate overall disagreement, neutral and overall agreement, from left to right respectively.
(TIFF)

**S1 Survey. Survey completed by practitioners.**
(PDF)

**S2 Survey. Survey completed by players.**
(PDF)

**S1 Table. Proportion and frequency of coach/support staff and player responses to awareness and experience of using VR.**
(DOCX)

## Acknowledgments

The authors would like to thank all practitioners and players who took part in this survey, as well as Dr John Perry for help with the PLS-SEM analysis.

## Author Contributions

**Conceptualization:** Ben Greenhough, Steve Barrett, Chris Towlson, Grant Abt.

**Data curation:** Ben Greenhough.

**Formal analysis:** Ben Greenhough.

**Investigation:** Ben Greenhough, Steve Barrett.

**Methodology:** Ben Greenhough, Steve Barrett, Chris Towlson, Grant Abt.

**Project administration:** Ben Greenhough.

**Resources:** Ben Greenhough.

**Software:** Ben Greenhough.

**Supervision:** Steve Barrett, Chris Towlson, Grant Abt.

**Validation:** Ben Greenhough, Steve Barrett, Chris Towlson, Grant Abt.

**Visualization:** Ben Greenhough.

**Writing – original draft:** Ben Greenhough.

**Writing – review & editing:** Ben Greenhough, Steve Barrett, Chris Towlson, Grant Abt.

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
