## [Decision Letter · Decision Letter 0]

19 Aug 2021

PONE-D-21-22184

Perceptions of professional soccer coaches and players toward virtual reality and the factors that modify their intention to use it.

PLOS ONE

Dear Dr. Greenhough,

Thank you for submitting your manuscript to PLOS ONE. After careful consideration, we feel that it has merit but does not fully meet PLOS ONE’s publication criteria as it currently stands. Therefore, we invite you to submit a revised version of the manuscript that addresses the points raised during the review process.

You will see that both reviewers are complimentary about your work but both offer suggestions relating to minor revisions in order to help strengthen your manuscript. 

We look forward to receiving your revised manuscript.

Kind regards,

Greg Wood, PhD

Academic Editor

PLOS ONE

Journal Requirements:

3. Please upload a copy of Supporting Information “S2 Text. Surveys’ definition of virtual reality” which you refer to in your text on page 24.

Reviewers' comments:

Reviewer's Responses to Questions

**Comments to the Author**

1. Is the manuscript technically sound, and do the data support the conclusions?

Reviewer #1: Yes

Reviewer #2: Yes

2. Has the statistical analysis been performed appropriately and rigorously? 

Reviewer #1: Yes

Reviewer #2: I Don't Know

3. Have the authors made all data underlying the findings in their manuscript fully available?

Reviewer #1: Yes

Reviewer #2: Yes

4. Is the manuscript presented in an intelligible fashion and written in standard English?

Reviewer #1: Yes

Reviewer #2: Yes

5. Review Comments to the Author

Reviewer #1: This is a nice study that describes the intention to use VR in high performance sport. To begin, I was sceptical believing that this type of study is necessary, being whether the content of this manuscript truly benefits/impacts the wider community in high performance sport and the opinions of VR. During the reading of the manuscript, I kept an open mind and tried not to bring my own beliefs of VR into my review. I believe that overall, it does contribute (albeit unfortunately not greatly in my opinion) to progressing the research area of VR. The writing was of great quality, the story/layout was very clear, and I believe that it is better to be published in an open access journal than not to be published at all– especially given the time point we are in within the Gartner Hype Cycle which I was not aware of beforehand but can completely relate to this in high performance sport environments.

After reading the manuscript over a few times, I feel that my comments are very generic guidelines that specific sentences or paragraphs. The authors are welcome to adjust their manuscript accordingly, or disagree with my comments too. As long as they are justified from the authors’ perspective.

1. The authors appear to be targeting this manuscript to staff in high-performance, but from my experience unfortunately I can imagine that they will most likely lose attention throughout reading the text. I understand that many of the data is necessary with the use of such questionnaire, but 6 tables, 5 figures and 6 supplementary files was a bit overwhelming. Are all these necessary including all of the text describing the findings too? It just seems that the manuscript only states facts, whereas there is not so much "why" attached to a lot of the "what".

2. This comment is once more related to the target audience of this manuscript. Is it possible that with the word count potentially shortened for the next draft that the authors could include a practical application/recommendations paragraph for practitioners who want to implement/improve adoption rate for VR into their club? For example, the authors could explain that focusing on the social influences of VR to get teams to use it is not the right approach (whereas the media department might favour this), but instead constructing a clear argument of VR for stakeholders in the club i) to support the physios training injured players with supported by recent research (Gokeler), ii) to support performance analysis department for tactical analysis and iii) collaborations with universities with VR technology that would loan out headsets to clubs in exchange for data – thereby reducing the cost barrier which directly solves the largest implementation barrier of VR.

3. One small thing: The title is slightly misrepresenting of the population of the sample. I wouldn’t use coaches and players, as in the manuscript there are many other support staff that are not coaches. I believe you’ve referred to them collectively as “practitioners” in L143.

Reviewer #2: This study examined attitudes towards the use of virtual reality in professional football. It is interesting and timely. I enjoyed reading this manuscript and have only minor comments.

The ethics statement mentions Declaration of Helsinki, but the latest version of this (item 35 in https://www.wma.net/policies-post/wma-declaration-of-helsinki-ethical-principles-for-medical-research-involving-human-subjects/) requires preregistration. As far as I can tell, this was not done by the authors, so they should at least mention which version of the Declaration their study adhered to (or remove this altogether). (Also on line 140-141.)

Abstract could be more informative about the findings and its implications.

Introduction: very clear motivation for the study – nothing to comment, really.

Line 165-167: Do you mean screen-based VR (e.g., CAVE-like set-ups) was not included? That needs to be made explicit throughout.

Line 189: a visual representation of the model would make sense for the less informed readers.

Power calculation/interpretation of p-values: was the amount of effects included in the model considered? Any regression based model with multiple predictor variables is influenced by issues of multiple comparison just as e.g., when performing multiple t-tests. Using alpha = 0.05 for individual predictor variables thus increases the chance of erroneously including these in the model (Type 1 errors). I would expect some treatment of this issue in the paper. See e.g., http://citeseerx.ist.psu.edu/viewdoc/download?doi=10.1.1.490.7640&rep=rep1&type=pdf

Line 230: how was this non-normality dealt with? Was normality not an assumption of the statistical model? This is important information since it could affect the validity of the statistical results/inferences.

Line 260: ‘strong theoretical support’ sounds somewhat vague.

The paper makes a distinction between potential cognitive, mental, tactical, physical, and technical benefits of VR. I think this needs to be defined very clearly in the Introduction. I would argue that a potential benefit (sensorimotor/visuomotor control, or something of this nature) is not perfectly captured by any of these terms, although I suspect the authors might capture this under ‘cognitive’ (which I think is suboptimal). Virtual reality could have training benefits in terms of an improved link between perception and action (e.g., better ability to link visual information about the kicker’s kinematics and ball flight in free kick or penalty scenarios to actions through VR training using a large database of kick kinematics).

Some readers may prefer a visual representation of the SEM – there are standardized ways to present these. In general, the current presentation of the model results is hard to follow (in terms of the interpretation/meaning of the results).

The discussion refers to the names of the factors in the model (‘performance expectancy’, ‘social influence’ etc), which sometimes affects readability. I would suggest writing out these in terms of the actual meaning to help the reader along. For example

“Our study allows us to conclude that performance expectancy is the largest contributor towards likeliness to use VR in professional soccer.”

->

“Our study allows us to conclude that likeliness to use VR in professional soccer mostly depends on

expectations concerning performance benefits of using VR.”

Signed

Joost Dessing

6. PLOS authors have the option to publish the peer review history of their article (what does this mean?). If published, this will include your full peer review and any attached files.

Reviewer #1: **Yes: **Adam Beavan

Reviewer #2: **Yes: **Joost Dessing

---

## [Author Response · Author response to Decision Letter 0]

23 Oct 2021

Edits have been made to the manuscript to meet the PLOS ONE style requirements as documented in the links above. 

Both surveys for coaches/support staff and players have been included in the supporting information as PDF files. 

3. Please upload a copy of Supporting Information “S2 Text. Surveys’ definition of virtual reality” which you refer to in your text on page 24.

I have subsequently removed “S2 Text. Survey definition of virtual reality” based on the comments I made back to reviewer 1 (see below). 

Reference list has been checked to ensure it is complete and correct. One new reference has been added (reference 30) to answer reviewer 2 comment. Reference has been included below 

Reviewer comments 1

After reading the manuscript over a few times, I feel that my comments are very generic guidelines that specific sentences or paragraphs. The authors are welcome to adjust their manuscript accordingly, or disagree with my comments too. As long as they are justified from the authors’ perspective.

1. The authors appear to be targeting this manuscript to staff in high-performance, but from my experience unfortunately I can imagine that they will most likely lose attention throughout reading the text. I understand that many of the data is necessary with the use of such questionnaire, but 6 tables, 5 figures and 6 supplementary files was a bit overwhelming. Are all these necessary including all of the text describing the findings too? It just seems that the manuscript only states facts, whereas there is not so much "why" attached to a lot of the "what".

We do agree with the reviewer’s comments. The manuscript is lengthy, with a lot of figures and tables. That said, this comment conflicts with reviewer 2’s comments, where they suggest adding two more figures below (diagrams of the SEM). As such, we have removed some supplementary information (S2, S3 and S4) as they are captured in the surveys (S5 and S5). We have removed the demographic text (original text lines 290-294) which is already captured in table 2. Table 3 has been moved to the supplementary materials (as its use in the manuscript is limited). We have also condensed the text between lines 311 – 364 (original text) to just a few of the keys findings, as this text is a repeat of what is shown in each figure. We have just added one of the SEM models (picture included for your reference at bottom of this file) that reviewer two has suggested. 

2. This comment is once more related to the target audience of this manuscript. Is it possible that with the word count potentially shortened for the next draft that the authors could include a practical application/recommendations paragraph for practitioners who want to implement/improve adoption rate for VR into their club? For example, the authors could explain that focusing on the social influences of VR to get teams to use it is not the right approach (whereas the media department might favour this), but instead constructing a clear argument of VR for stakeholders in the club i) to support the physios training injured players with supported by recent research (Gokeler), ii) to support performance analysis department for tactical analysis and iii) collaborations with universities with VR technology that would loan out headsets to clubs in exchange for data – thereby reducing the cost barrier which directly solves the largest implementation barrier of VR.

We have included a practical applications paragraph after the conclusions, which does help to summarise some of the take-home messages of the manuscript. Thank you for the suggestion. 

3. One small thing: The title is slightly misrepresenting of the population of the sample. I wouldn’t use coaches and players, as in the manuscript there are many other support staff that are not coaches. I believe you’ve referred to them collectively as “practitioners” in L143.

We have included ‘support staff’ in the title to represent a large proportion of the sample that are not coaches. We then refer to them as practitioners in the methods and then throughout the manuscript to allow for better reading.

Reviewer 2 comments 

The ethics statement mentions Declaration of Helsinki, but the latest version of this (item 35 in https://www.wma.net/policies-post/wma-declaration-of-helsinki-ethical-principles-for-medical-research-involving-human-subjects/) requires preregistration. As far as I can tell, this was not done by the authors, so they should at least mention which version of the Declaration their study adhered to (or remove this altogether). (Also on line 140-141.)

Thank you for providing this information. We weren’t aware of the need to preregister the study to include the Declaration of Helsinki. This study is not preregistered and so we have removed this part of the ethics statement. 

Abstract could be more informative about the findings and its implications.

We feel that the abstract adequately conveys the aim, method, results summary, and main implications. However, we have made some minor adjustments to aid clarity. 

Introduction: very clear motivation for the study – nothing to comment, really.

Thank you 

Line 165-167: Do you mean screen-based VR (e.g., CAVE-like set-ups) was not included? That needs to be made explicit throughout.

Yes, CAVE-like set ups were not included as part of our definition of VR. We have included a sentence on line 186 to make this explicit, along with a reference for readers to learn more about CAVE set ups. 

Line 189: a visual representation of the model would make sense for the less informed readers.

We have included a visual representation of the model in Figure 6 (line 485).

Power calculation/interpretation of p-values: was the amount of effects included in the model considered? Any regression based model with multiple predictor variables is influenced by issues of multiple comparison just as e.g., when performing multiple t-tests. Using alpha = 0.05 for individual predictor variables thus increases the chance of erroneously including these in the model (Type 1 errors). I would expect some treatment of this issue in the paper. See e.g., http://citeseerx.ist.psu.edu/viewdoc/download?doi=10.1.1.490.7640&rep=rep1&type=pdf

As outlined by Henseler et al (2014) the bias-corrected and accelerated confidence intervals (BCa) bootstrapping method as implemented in SmartPLS maintains the type 1 error rate at 5%. We have added that clarification to line 275-278. 

Henseler, J., Dijkstra, T. K., Sarstedt, M., Ringle, C. M., Diamantopoulos, A., Straub, D. W., Ketchen, D. J., Hair, J. F., Hult, G. T. M., & Calantone, R. J. (2014). Common Beliefs and Reality About PLS. Organizational Research Methods, 17(2), 182–209. https://doi.org/10.1177/1094428114526928

Line 230: how was this non-normality dealt with? Was normality not an assumption of the statistical model? This is important information since it could affect the validity of the statistical results/inferences.

Normality is not an assumption with PLS-SEM. We have included a sentence and reference within the revised manuscript (line 251). 

Line 260: ‘strong theoretical support’ sounds somewhat vague.

We have edited this sentence from “required strong theoretical support for its inclusion in the formative construct model” to required (1) a rationale for its inclusion in the formative construct model through either anecdotal or empirical evidence [29], and (2) removing the indicator variable doesn’t change the conceptual meaning of the construct [30] (lines 286-288). 

The paper makes a distinction between potential cognitive, mental, tactical, physical, and technical benefits of VR. I think this needs to be defined very clearly in the Introduction. I would argue that a potential benefit (sensorimotor/visuomotor control, or something of this nature) is not perfectly captured by any of these terms, although I suspect the authors might capture this under ‘cognitive’ (which I think is suboptimal). Virtual reality could have training benefits in terms of an improved link between perception and action (e.g., better ability to link visual information about the kicker’s kinematics and ball flight in free kick or penalty scenarios to actions through VR training using a large database of kick kinematics).

In the Introduction we already refer to perceptual motor-skills (line 116), and studies related to this. We feel that this clearly highlights to the reader that perception/action is a potential benefit of VR. 

Some readers may prefer a visual representation of the SEM – there are standardized ways to present these. In general, the current presentation of the model results is hard to follow (in terms of the interpretation/meaning of the results).

We have now added a figure (Figure 6) to visually represent the SEM model (line 485).

The discussion refers to the names of the factors in the model (‘performance expectancy’, ‘social influence’ etc), which sometimes affects readability. I would suggest writing out these in terms of the actual meaning to help the reader along. For example

“Our study allows us to conclude that performance expectancy is the largest contributor towards likeliness to use VR in professional soccer.”

->

“Our study allows us to conclude that likeliness to use VR in professional soccer mostly depends on

expectations concerning performance benefits of using VR.”

Thank you for that suggestion. We have made edits to the discussion/conclusion to improve the readability.

---

## [Decision Letter · Decision Letter 1]

1 Dec 2021

Perceptions of professional soccer coaches, support staff and players toward virtual reality and the factors that modify their intention to use it.

PONE-D-21-22184R1

Dear Dr. Greenhough,

We’re pleased to inform you that your manuscript has been judged scientifically suitable for publication and will be formally accepted for publication once it meets all outstanding technical requirements.

Kind regards,

Greg Wood, PhD

Academic Editor

PLOS ONE

Additional Editor Comments (optional):

Reviewers' comments:

Reviewer's Responses to Questions

**Comments to the Author**

1. If the authors have adequately addressed your comments raised in a previous round of review and you feel that this manuscript is now acceptable for publication, you may indicate that here to bypass the “Comments to the Author” section, enter your conflict of interest statement in the “Confidential to Editor” section, and submit your "Accept" recommendation.

Reviewer #1: All comments have been addressed

Reviewer #2: All comments have been addressed

2. Is the manuscript technically sound, and do the data support the conclusions?

Reviewer #1: Partly

Reviewer #2: Yes

3. Has the statistical analysis been performed appropriately and rigorously? 

Reviewer #1: Yes

Reviewer #2: Yes

4. Have the authors made all data underlying the findings in their manuscript fully available?

Reviewer #1: Yes

Reviewer #2: Yes

5. Is the manuscript presented in an intelligible fashion and written in standard English?

Reviewer #1: Yes

Reviewer #2: Yes

6. Review Comments to the Author

Reviewer #1: Dear authors of this manuscript, thank you for replying to the manuscript. I have decided to accept the manuscript on the basis that it may benefit practitioners in the wider community and push the use of VR further in sporting organisations being an open access journal. A few recommendations about addressing the reviewer comments: There is no need to show in so much detail what changes were made. For example, keeping all of the 'track changes' comments on just creates a bit of mess for us reviewers to sift through. I recommend deleting these formatting comments made before resubmitting the revisions for next time. For example, see Figure 6 with all of the font/bold/italic comments that aren't necessary to keep in a review. Also, when you delete small words throughout the manuscript, you can "accept" them on track changes to make them disappear, while keeping the new word in still in red font. For example, see line 244-245, it's unnecessarily messy to read. Another nitpicking comment would be the less than adequate responses to the reviewer comments. For example, I feel Reviewer 2 made a good point about: "The paper makes a distinction between potential cognitive, mental, tactical, physical, and technical benefits of VR. I think this needs to be defined very clearly in the Introduction." But the response was rather underwhelming. Authors are welcome to defend what they have written and disagree with the reviewer, but this has to be better justified. Simply putting that "We feel that this clearly highlights to the reader that perception/action is a potential benefit of VR" does not adequately address that reviewer feels that it doesn't. More effort should be made to better support why the reviewer may have not seen eye-to-eye with the authors, and try and guide them in a more effortful reply to overcome the differences in opinions.

Reviewer #2: Apologies for the lateness of the review. I have been stomped with marking.

I am happy/satisfied with the changes and have nothing more to add, other than to suggest doing a quick check for grammar and spelling - I noticed a few mistakes.

7. PLOS authors have the option to publish the peer review history of their article (what does this mean?). If published, this will include your full peer review and any attached files.

Reviewer #1: No

Reviewer #2: **Yes: **Joost Dessing

---

## [Editor Report · Acceptance letter]

20 Dec 2021

PONE-D-21-22184R1 

Perceptions of professional soccer coaches, support staff and players toward virtual reality and the factors that modify their intention to use it. 

Dear Dr. Greenhough:

I'm pleased to inform you that your manuscript has been deemed suitable for publication in PLOS ONE. Congratulations! Your manuscript is now with our production department. 

Kind regards, 

on behalf of

Dr. Greg Wood 

Academic Editor

PLOS ONE